# Meta-Analysis of the Effect of Ventilation on Intellectual Productivity

**DOI:** 10.3390/ijerph20085576

**Published:** 2023-04-19

**Authors:** Hayata Kuramochi, Ryuta Tsurumi, Yoshiki Ishibashi

**Affiliations:** 1Formerly of Faculty of Law, The University of Tokyo, Tokyo 113-0033, Japan; 2Nikken Sekkei Research Institute, Tokyo 101-0052, Japan; 3Department of Preventive Medicine and Public Health, Keio University School of Medicine, Tokyo 160-8582, Japan

**Keywords:** productivity, ventilation, indoor air quality, school, meta-analysis, systematic-review

## Abstract

Indoor air quality (IAQ) influences the health and intellectual productivity of occupants. This paper summarizes studies investigating the relationship between intellectual productivity and IAQ with varying ventilation rates. We conducted a meta-analysis of five studies, with a total of 3679 participants, and performed subgroup analyses (arithmetic, verbal comprehension, and cognitive ability) based on the type of academic performance. The task performance speed and error rate were evaluated to measure intellectual productivity. The effect size of each study was evaluated using the standardized mean difference (SMD). In addition, we calculated a dose-response relationship between ventilation rate and intellectual productivity. The results show that the task performance speed improved, SMD: 0.18 (95% CI: 0.10–0.26), and the error rate decreased, SMD: −0.05 (95% CI: −0.11–0.00), with an increase in ventilation rate. Converting the intervention effect size on the SMD into the natural units of the outcome measure, our analyses show significant improvements in the task performance speed: 13.7% (95% CI: 6.2–20.5%) and 3.5% (95% CI: 0.9–6.1%) in terms of arithmetic tasks and cognitive ability, respectively. The error rate decreased by −16.1% (95% CI: −30.8–0%) in arithmetic tasks. These results suggest that adequate ventilation is necessary for good performance.

## 1. Introduction

People are exposed to air pollutants in their daily lives, and various adverse health effects are of concern. In particular, because 90% of our time is spent indoors [1], the green building concept, which aims to bring direct health benefits to occupants by improving the quality of the indoor environment, is gaining importance in public and environmental health [2]. Indoor air quality (IAQ) is an important criterion, as the concentration of many airborne pollutants is higher indoors than outdoors [3].

Furthermore, IAQ has been noted in several studies to affect human intellectual productivity [4,5,6]. For example, Federspiel et al. [7] and Shendell et al. [8] showed that the magnitude of the difference between indoor and outdoor carbon dioxide (CO_2_) concentrations is associated with poor work performance and increased student absenteeism. In addition, several studies have indicated that levels of CO_2_ that pose no health risks, such as around 1000 ppm, can still have an impact on intellectual productivity [5,9].

Indoor CO_2_ concentration is often used as an overall indicator of IAQ owing to its ease of measurement compared to that of other toxins [10]. On the other hand, it has been reported that indoor toxic substances other than CO_2_ can also affect intellectual productivity [11,12]. For example, it has been noted that test scores decrease as the concentration of VOCs and PM2.5 increases [13,14]. However, most studies have set CO_2_ concentration as an independent variable owing to its ease of measurement compared to that of other toxic substances [10]. Among the airborne toxins that may affect the performance of occupants, CO_2_ is the only human-derived pollutant (unless there is combustion in the room). Conversely, VOCs and other harmful substances apart from CO_2_ are derived from building materials. In a study focusing on CO_2_, IAQ may affect human cognitive ability; however, this may be overlooked when the CO_2_ concentration is low but VOCs are high. Therefore, by setting the ventilation rate as an independent variable to comprehensively evaluate the effect of all harmful substances in the air, we can analyze the impact of IAQ on intellectual productivity more precisely. There are independent studies on the effect ventilation rate changes on intellectual productivity [15,16]. However, the outcomes of these studies are not consistent, perhaps because they differ in study design and target population. Although literature reviews on exposure to CO_2_ exist [10,17,18], a meta-analysis of the experimental results on ventilation has not been performed to date.

Therefore, this study reports the results of statistical estimation of the effect of ventilation on intellectual productivity using a meta-analysis to organize the evidence of the relationship between ventilation rate and intellectual productivity. In addition, this study provides evidence that could be used as guidelines to improve intellectual productivity indoors.

## 2. Methods

### 2.1. Search Strategy

This study conducted a meta-analysis according to the PRISMA [19] statement and MOOSE [20], which are guidelines for meta-analysis. This paper was checked against the PRISMA checklist (Appendix A) and is not registered on PROSPERO. The research questions, study selection criteria, and eligibility criteria were determined in advance by the research group to eliminate bias in the collection of literature. The research question was defined as “whether the change in ventilation rate affects intellectual productivity.” The conditions for extracting literature from the database (PICO) as a research selection criterion are as follows:Patient: General population, such as students or office workersIntervention: Change in ventilation rateControl: No interventionOutcome: Intelligent productivity

As the eligibility criteria, the paper must be an accessible English-language paper that has been peer-reviewed and published in a journal by 31 August 2020, and the intellectual productivity must be assessed by at least one objectively measurable indicator.

In addition, as this study focuses on providing a basis for teachers, administrators, and industrial physicians to discuss the need for ventilation in schools and workplaces, we excluded laboratory experiments and observational studies. In the study of Wyon and Wargocki [21], it is stated that because participants in laboratory studies, with paid employees, tend to exert more effort than usual if the exposure time to the poor environment is short, the adverse effects observed in laboratory experiments may be underestimated compared to field studies. In addition, it is challenging to provide controlled and uniform interventions in actual scenarios, as opposed to laboratory experiments. For example, there have been cases where the results of interventions that were strictly controlled in a hospital setting could not be replicated in actual interventions in the field, thus emphasizing the importance of reporting research in the natural world in the field of clinical epidemiology [22]. Moreover, observational studies with CO_2_ concentration and ventilation rate as independent variables are not adjusted for other variables that are not of interest within the research; however, these variables may affect the results, such as the environmental conditions of the classroom where the observation occurred [23].

We used PubMed, Web of Science, and Scopus as the search databases. The search formula was determined after consultation with the librarians at Komaba Library, University of Tokyo, Japan, to satisfy PICO. The details of the search strategy are described in the Appendix A. Additional literature was evaluated by manually searching the literature cited by relevant articles and the list of literature that cited relevant reports.

### 2.2. Data Extraction and Quality Assessment

Literature searching was performed in two stages by two authors over a one-month period, starting in December 2019 (Figure 1). In the primary screening, we checked the titles and abstracts of the literature detected by the search to exclude those that did not conform to PICO. Those that could not be confirmed from the abstracts alone were retained. In the secondary screening, the text of the literature that passed the primary screening was reviewed, and those that met PICO and the eligibility criteria were retained. The screening results were then cross-checked between the two authors. The titles accepted by both authors were adopted, while those rejected by both authors were excluded. The literature accepted by one author but rejected by the other was individually evaluated to determine whether it was acceptable in consultation with a third author.

In the screening process, we rejected literature that did not report the ventilation rate for each group. For articles that lacked the detailed data required for meta-analysis (95% confidence interval (CI) or standard error), we attempted to obtain the data by contacting the authors of the articles via email. We incorporated the available data of titles whose authors responded.

As a result of preliminary research, the adaptable literature in this study was expected to be mostly crossover designs and minimal randomized controlled trials. Therefore, we utilized the risk of bias assessment used by Stieb et al. [24] to assess the quality of the study.

### 2.3. Data Synthesis and Analysis

In the initial literature review, two outcomes of intellectual productivity were measured: the task performance speed and the error rate. First, we conducted meta-analyses on the task performance speed and error rate as the main analysis in this study. Second, we conducted meta-analyses using measures of academic performance as a subgroup analysis. In classifying the subgroups, Haverinen et al. [16], who analyzed the relationship between IAQ and academic performance, characterized student performance using cognitive tests, intellectual productivity (e.g., task completion speed), and numerical or verbal tasks. Therefore, we decided to conduct a meta-analysis on six subgroups in this study by dividing them into three categories based on the type of task, namely, arithmetic, verbal comprehension, and cognitive ability tasks. The subgroups were created and classified based on two productivity indicators: the task performance speed and the error rate. We did not conduct a subgroup analysis based on each population characteristic because the adopted literature only included students. Third, to determine the relationship between the amount of improvement in academic performance and ventilation rate, dose-response analyses were performed [10]. As in the meta-analysis, the same two outcomes were examined. The specific methods are described in the Appendix A.

The standardized mean difference (SMD) between groups was adopted as a data integration method in the meta-analysis [25]. A standard random-effects model was used for the calculations [26]. Some of the data from the literature were converted into a form that could be used in the meta-analysis. Details on this conversion, including the equations used, can be found in the Appendix A. In the meta-analysis, it is difficult to understand the exact amount of effect of the intervention using the SMD. Therefore, a restatement of outcomes (RES) with actual units was performed using the study with the best risk of bias assessment as the baseline [25]. The RES details can be found in the Appendix A. Heterogeneity was evaluated using I^2^ (I-statistic). In addition, funnel plots were created to assess the presence of publication bias. In this study, a significance level of 5% or less was considered significant. All analyses were conducted using the statistical analysis language R (ver. 4.0.2), meta and metacor package [27].

## 3. Results

### 3.1. Results of Meta-Analysis

Five studies, 47 experiments, and a total of 3679 subjects were included in this meta-analysis of the relationship between ventilation and intellectual productivity. A summary of the adopted literature is listed in Table 1. In the studies adopted in this literature, ventilation rates ranged from 1.4–5.7 L/(s · person) in non-intervention and 6.5–9.9 L/(s · person) in intervention. The quality of each study assessed is listed in Table 2. The exposure assessment was high in all studies; however, the other items were low, with variation depending on whether the study designs were double-blind or not in the blinding item. The rationale for evaluating each item is listed in Appendix A. The classification of each test into subgroups is listed in Appendix A.

Figure 2 shows the integration of the task performance speed and the error rate for all studies. The increase in ventilation rate was significantly associated with the improvement of the task performance speed (SMD: 0.20, 95% CI: 0.12–0.28, *p* < 0.01, I^2^ = 84%), and the error rate tended to decrease (SMD: −0.05, 95% CI: −0.09–0.00, *p* = 0.04, I^2^ = 55%). Regarding heterogeneity, high heterogeneity was found for the task performance speed and moderate heterogeneity was found for the error rate. Although each study’s direction of point estimates on the task performance speed differed, the confidence intervals overlapped. Thus, the inconsistency was not severe for both plots.

### 3.2. Subgroup Analysis

Figure 3 shows the forest plot of the combined experiments for the arithmetic, verbal comprehension, and cognitive ability tasks. The results for the task performance speed for arithmetic, verbal comprehension, and cognitive ability tasks were as follows: (SMD: 0.24, 95% CI: 0.12–0.36, RES: 13.7%, 95% CI: 6.2–20.5%, *p* < 0.01, I^2^ = 82%), (SMD: 0.16, 95% CI: −0.03–0.36, RES: 8.9%, 95% CI: −1.8–19%, *p* = 0.10, I^2^ = 93%), and (SMD: 0.16, 95% CI 0.10–0.23, RES: 3.5%, 95% CI: 0.9–6.1%, *p* < 0.01, I^2^ = 0%), respectively. The increased ventilation increased the task performance speed in all subgroups, and the effects were significant in the arithmetic and cognitive ability tasks. In addition, the error rates decreased in the arithmetic (SMD: −0.10, 95% CI: −0.18–−0.03, RES: −16.1%, 95% CI: −30.8–0%, *p* = 0.01, I^2^ = 52%) and verbal comprehension tasks (SMD: −0.03, 95% CI: −0.08–0.01, RES: −5.3%, 95% CI: −21.1–8.8%, *p* = 0.15, I^2^ = 1%); however, only that of the arithmetic task was significant. No association was found for the error rate in cognitive ability task (SMD: 0.02, 95% CI: −0.11–0.15, RES: 0%, 95% CI: −0.9–0.9%, *p* = 0.76, I^2^ = 74%). A high degree of heterogeneity was found in the task performance speed of these arithmetic and verbal comprehension tests and in the error rate of the cognitive ability tests. In the task performance speed for the arithmetic task, the directions of the point estimates in each study were not different, and the confidence intervals overlapped; therefore, the inconsistency was not important. In the task performance speed for the verbal comprehension tasks, the directions of the point estimates in each study were different, and the confidence intervals did not overlap; therefore, the inconsistency might be important. In the error rate for the cognitive ability task, the directions of the point estimates in each study were different, but the confidence intervals overlapped; therefore, the inconsistency might be important.

A moderate degree of heterogeneity was found in the error rate of the arithmetic tests. Although the directions of the point estimates in each study were different, the confidence intervals overlapped; therefore, the inconsistency was not important. No heterogeneity was found in the other subgroups.

### 3.3. Dose-Response Analysis

The results of the dose-response analyses between the ventilation rate and the task performance speed and the error rate based on the experimental data from the literature adopted in this study are shown in Figure 4. The horizontal axis shows the mean of the ventilation rate in the control and intervention groups, and the vertical axis shows the variation in the task performance speed and error rate when the ventilation rate is increased by 1 L/(s · person) over $VR_{mid}$ in the form of SMDs. The effect of increased ventilation was expected to be reduced as the pre-intervention ventilation level increased. For the task performance speed, if the diminishing effect is assumed to be linear, the intellectual productivity improvement from increased ventilation tended to diminish with the higher ventilation rate before intervention; however, the tendency was not significant (*p* = 0.09). It is estimated that the improvement due to increased ventilation disappears at 10.7 L/(s · person). Conversely, the higher ventilation rate before intervention was expected to reduce the improvement in intellectual productivity from the increased ventilation in the error rate; however, this tendency was not found (*p* = 0.27).

### 3.4. Evaluation of Publication Bias

We created funnel plots to determine the effect of publication bias in the search and acceptance of studies. As shown in Figure 5, no significant bias was found, and no evidence of publication bias was found in this study.

## 4. Discussion

We conducted meta-analyses using experimental results from five studies, with a total of 3679 participants, to discover the effect of ventilation on intellectual productivity. Overall, our results suggest that an increased ventilation rate is significantly associated with improved task performance speed and, although not significant, also tend to improve the error rate (Figure 2). The association between increased ventilation and enhanced task performance speed shown in this study is consistent with previous studies. Several studies have reported an association between reduced CO_2_, one of the most well-known toxins that are removed by ventilation, and increased productivity [15,33]. For example, Bakó-Biró et al. [15] found that task performance of cognitive skill improved by 2.2–15% with decreasing CO_2_ concentration. Some of our analyses showed an improvement of 13.7%, which is close to the maximum reported in the above study. These outcomes suggest that factors affecting performance might include not only CO_2_ but also other toxic substances, such as VOCs. However, CO_2_ and VOCs may vary independently under insufficient ventilation conditions. Therefore, indirect studies and studies that only focus on CO_2_ may not determine the effect of VOCs. Our analysis was conducted on intervention studies of ventilation only, which allowed us to fully follow the impact of the air conditions on occupants.

There are several possible mechanistic explanations for the association between increased ventilation and improved task performance speed identified in this analysis. Maddalena et al. [34] reported a significant decrease in decision-making performance with decreased ventilation. They observed that the concentrations of CO_2_ and VOCs increased simultaneously with a decrease in ventilation. This outcome suggests that an increased ventilation rate improves the overall IAQ, including CO_2_, VOCs, and other toxic substances, and that the shift in the environment may have improved performance. Jacobson et al. [18] found that exposure to a typical CO_2_ concentration in an indoor environment (<5000 ppm) caused a decline in cognitive skill through autonomic nervous system changes, such as increased heart rate, leading to acute health symptoms, such as headache and drowsiness. While previous studies focused on the relationship between CO_2_ concentration and performance, our study quantitatively assessed the relationship between IAQ improvement and performance, suggesting the need to investigate the mechanism of performance decline with an increase in the concentration of toxic substances other than CO_2_, such as VOCs. In considering the mechanisms behind the impact of CO_2_ and VOC on intellectual productivity, it is desirable to note the potential effects of combined exposure and to comprehensively measure and analyze substances that may affect intellectual productivity. Therefore, we anticipate future studies that will address these concerns.

In the subgroup analysis, the increase in the task performance speed for the arithmetic and verbal comprehension tasks showed a significant improvement in the RES. In contrast, the task performance speed for the cognitive skill task was significant; however, the RES was small (Figure 3). The questions in the arithmetic and verbal comprehension tests required complex processing, such as arithmetic operations and proofreading, while the cognitive skill tests required simple processing, such as reflexes and comparing the shapes of numbers. The difference between the improvement in the task performance speed of the arithmetic and verbal comprehension tests and the cognitive ability tests is in agreement with Wargocki and Wyon [19], who suggested that the improvement in the ventilation rate is greater when solving more complex problems. A possible mechanism for this is that the mental load felt by the test participants was different when solving tasks of different difficulty levels, which may have resulted in a smaller improvement in ventilation [17]. Du et al. [17] stated that if the mental effort of the test participant exceeds the mental load of the test, then the effect of CO_2_ on cognitive performance may be difficult to determine. Thus, the difficulty of the task may be related to the ease with which IAQ affects the room occupants. This suggests that the implementation of ventilation in workplaces where occupants perform complex intellectual tasks is more effective than in workplaces where occupants perform simple tasks. Further research on the effect of increased ventilation on the improvement in complicated tasks is needed. The verbal comprehension task performance speed was the most heterogeneous. This seems to be caused by the difficulty of the verbal comprehension test, which may also have increased the variance.

Furthermore, the dose-response analyses suggested that the increase of the ventilation rate was more effective in improving the task performance speed when the ventilation rate was lower (Figure 4). In a classroom with 10–12 year-old students, the ventilation requirement is 6.71 L/(s · person) based on ASHRAE 62.1 (ASHRAE, American Society of Heating, Refrigerating and Air-Conditioning Engineers) [35]. Similarly, according to the method based on perceived air quality in EN 16798 (EN, Europe norm), the ventilation requirement for category II (less than 20% of people are unsatisfied) is 9.8 L/(s · person) [36]. In contrast, the upper limit of the ventilation rate required in our study to improve the task performance speed is 10.7 L/(s · person). This outcome is close to EN 16798. Thus, EN 16798 is reasonable in terms of intellectual productivity. Our result could be used in the architectural design and environmental management of buildings.

This study has potential limitations. It should be noted that the conditions under which the studies were conducted are similar. As listed in Table 1, four out of five of the adopted studies were in Denmark, and the participants in these studies were 10–12 years of age. Since only English language literature was included, there is a possibility of bias due to the language of the published papers. Additionally, the absence of PROSPERO registration raises the possibility of selective reporting and outcome reporting bias. Therefore, caution should be exercised in applying the results of this meta-analysis to other conditions. In addition, the meta-analysis in this literature did not reflect biases in the IAQ measurement and the research methods in each study. The meta-analysis was conducted without considering the risk of bias assessment and other biases in the IAQ measurement in each study. Consequently, the results could possibly be not significantly consistent in one direction.

In the studies adopted in this literature, the ventilation rate ranged from 1.4 to 5.7 L/(s · person) in non-intervention and from 6.5 to 9.9 L/(s · person) in intervention. The CO_2_ concentrations ranged from 952 to 4140 ppm in the non-intervention period and from 501 to 983 ppm in the intervention period. Further studies are required to investigate the effects on intellectual productivity over a broader range of ventilation rates.

## 5. Conclusions

The most important finding of this study is that the effect of ventilation on one aspect of intellectual productivity was quantitatively demonstrated. In particular, the results showed the usefulness of ventilation in terms of the task performance speed and error rate. In the dose-response analysis, the positive relationship between ventilation rate and the task performance speed was no longer observed at 10.7 L/(s · person).

The findings of this study will be helpful for policy decisions on indoor environmental management. In addition, the results indicate that intellectual productivity should also be taken into account when determining ventilation standards. In the future, it is desirable to establish further evidence of the relationship between IAQ and intellectual productivity.

## Figures and Tables

**Figure 1 ijerph-20-05576-f001:**
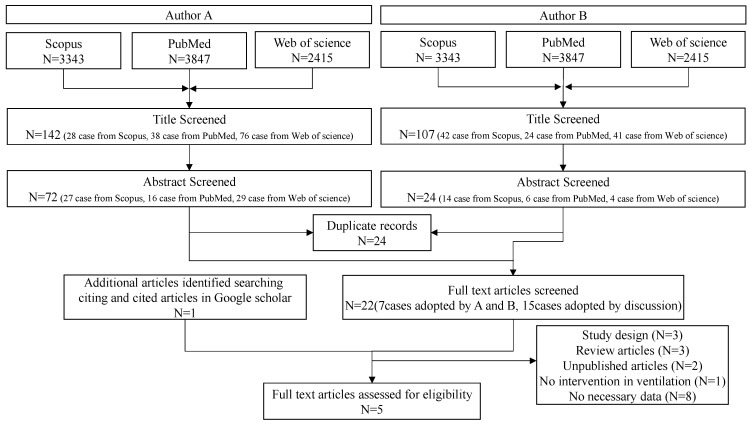
Screening flow diagram of the search strategy.

**Figure 2 ijerph-20-05576-f002:**
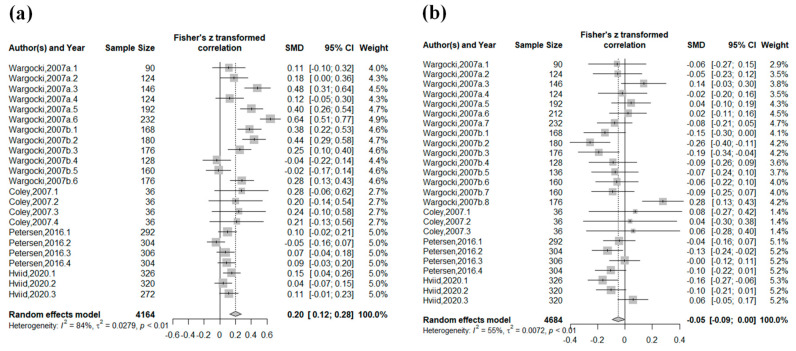
Association between the increase in ventilation rates and performance of school tests: (**a**) the task performance speed; and (**b**) the error rate.

**Figure 3 ijerph-20-05576-f003:**
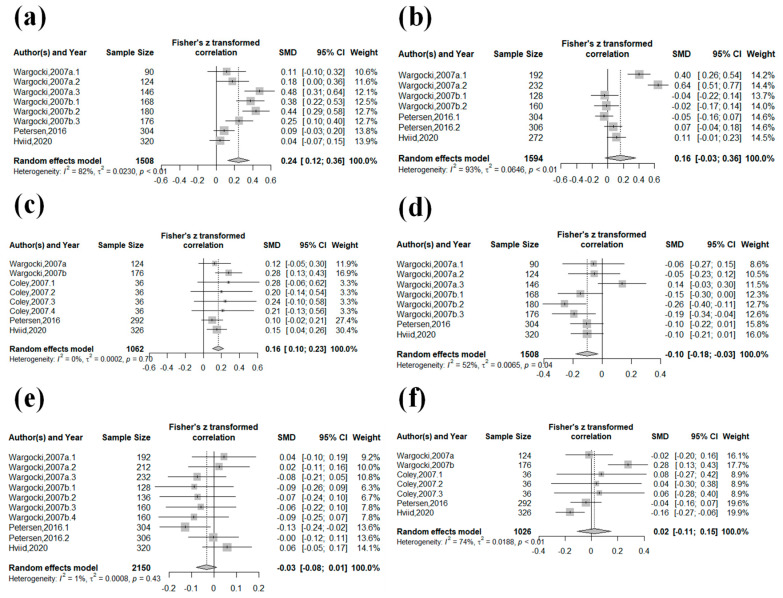
Association between the increase in ventilation rate and the performance of school tests according to subgroups: (**a**) arithmetic, the task performance speed; (**b**) arithmetic, the error rate; (**c**) verbal comprehension, the task performance speed; (**d**) verbal comprehension, the error rate; (**e**) cognitive skill, the task performance speed; and (**f**) cognitive skill, the error rate.

**Figure 4 ijerph-20-05576-f004:**
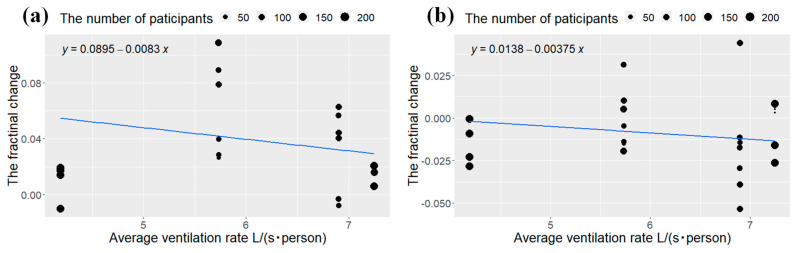
Dose-response relationship between ventilation rate and performance of intelligent tests: (**a**) the task performance speed; and (**b**) the error rate. Each point in these figures represent the change in performance with an increase in ventilation rate of 1 L/(s · person) from the average ventilation rate. The size of these points depends on the sample size of the studies. The blue line is a regression line.

**Figure 5 ijerph-20-05576-f005:**
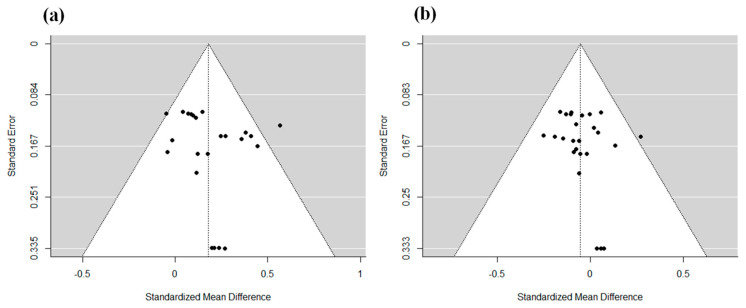
Funnel plot for publication bias: (**a**) speed of answer and (**b**) the error of rate. Each study in the analyses is plotted as a black dot. The vertical lines are the results of the meta-analyses.

**Table 1 ijerph-20-05576-t001:** Summary of studies included in the meta-analysis. ^†^ Reported as average concentration (range).

Lead Author (Publication Year)	Study Design	Location	Season	Population Source	Number of Participants	Age of Participants	Ventilation Type	Method of Estimating Ventilation Rates	Range of Estimated Ventilation Rate L/(s · person)	The Presence or Absence of Room Temperature Control	Measured CO_2_ Concentration ^†^	Learning Outcome
Wargocki and Wyon (2007) [28]	2 × 2 crossover intervention study	Denmark	Winter (January)	2 classrooms in 1 elementary school	44	10–12	Mechanical ventilation	Calculation by measured CO_2_ concentrations and occupant density	3–9.5	The teachers were free to alter the thermostatic valves on the radiators at any time.	1102 ppm (925–1280)	School tasks
Wargocki and Wyon (2007) [29]	2 × 2 crossover intervention study	Denmark	Late summer (August, September)	2 classrooms in 1 elementary school	44	10–12	Mechanical ventilation	Calculation by measured CO_2_ concentrations and occupant density	2.7–9.9	Air temperature was manipulated by either operating or idling split cooling units.	888 ppm (775–1000)	School tasks
Coley et al. (2007) [30]	Crossover intervention study	England	Summer	1 classroom in 1 elementary school	18	10–11	Natural ventilation	Calculation by measured CO_2_ concentrations and occupant density	1.5–13	Air temperature was maintained by the use of a freestanding air conditioning unit.	1800 ppm (700–2900)	Psychological tests
Petersen et al. (2016) [31]	Double-blind crossover intervention study	Denmark	Autumn	1 classroom in 1 elementary school	13–24	10–12	Mechanical ventilation	Calculation by measured CO_2_ concentrations and occupant density	1.4–6.6	Ventilation units were equipped with an electrical heating coil.	1205 ppm (800–1610)	Academic performance tests
Hviid et al. (2020) [32]	Double-blind 2 × 2 crossover intervention study	Denmark	Autumn (August, September)	4 classrooms in 1 elementary school	23	10–12	Mechanical ventilation	Calculation by measured CO_2_ concentrations and occupant density	3.9–10.6	Ventilation supply temperature was controlled according to ventilation rate	1183 ppm (718–1648)	Cognitive performance test

**Table 2 ijerph-20-05576-t002:** Summary of risk of bias ratings. The assessment was based on Stieb et al. [24] criteria. Reasons for these risk of bias ratings for individual studies are noted in Appendix A.

Lead Author (Publication Year)	Exposure Assessment	Outcome Assessment	Confounding	Completeness of Outcome Data	Selective Outcome Reporting	Conflicts of Interest	Other
Wargocki and Wyon (2007) [28]	Probably high	Probably low	Low	Low	Low	Probably low	Low
Wargocki and Wyon (2007) [29]	Probably high	Probably low	Low	Low	Low	Probably low	Low
Coley et al. (2007) [30]	Probably high	Probably low	Probably high	Low	Low	Probably low	Low
Petersen et al. (2016) [31]	Probably high	Probably low	Low	Low	Low	Probably low	Low
Hviid et al. (2020) [32]	Probably high	Probably low	Low	Low	Low	Low	Low

## Data Availability

The data presented in this study are available upon request to the corresponding author.

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
