# Peer review of "Meta-Analysis of the Effect of Ventilation on Intellectual Productivity"

_ijerph, 2023, doi:10.3390/ijerph20085576_

Round 1

Reviewer 1 Report

Why the error rate improvement has a negative sign in line 23?

Poor English in lines 42 to 46 was detected. The paragraph is incomprehensible and needs to be improved.

Author Response

Dear reviewer 1,

We thank reviewer 1 for making time available to this paper. We appreciate very much for his/her pointing out a few potentially improvement.

Kind regards,

Reviewer 2 Report

This paper is a meta-analysis of the effect of ventilation on cognitive function.   After a systematic search of the literature, the researchers found 5 papers published before August 31, 2020 that met their criteria, which they discussed and analyzed.

The researchers considered room CO2 levels to be the variable strongly affected by changes in ventilation.  They also mentioned VOCs, but they noted that levels of VOCs might change independently of CO2, when levels of ventilation are insufficient.

The one variable that the did not discuss is room temperature.  In Table 1 they noted that in 3 of the 5 studies they considered, room temperature control was absent.  I’m not sure if that means that the temperature control was not linked to the ventilation system – this should be clarified.  However, since overly warm room temperatures also impair cognitive function, and since changing ventilation often helps moderate room temperature, this topic should be included in the discussion.

Minor Error  Lines 38 & 39:  In addition, Satish et al.[5] and Wargocki and Wyon[9] suggest that CO2 concentrations below 2500 ppm may reduce cognitive performance.

I assume that you mean *above 2500*.

Author Response

Dear reviewer 2,

We thank reviewer 2 for making time available to this paper. We appreciate very much for his/her pointing out a few potentially improvement.

Kind regards,

Reviewer 3 Report

The authors have conducted commendable scientific research on the relationship between ventilation and intellectual productivity, effectively demonstrating the quantitative impact of ventilation on task performance speed. However, to further enhance the study's significance, I would appreciate additional clarification and suggestions for improvement, as the followings: 

1. Introduction is of good quality as it provides relevant and reliable information on the topic of indoor air quality and its effects on human health and productivity. It highlights the importance of the green building concept and its role in improving the quality of indoor environments to benefit occupants' health. Authors cite several studies that demonstrate the negative impact of indoor air pollution on human health and cognitive performance. The use of CO2 concentration as an overall indicator of IAQ is well-supported and explained.

2. Methods described by authors appear to be thorough and well-planned. The study followed the PRISMA and MOOSE guidelines for meta-analysis, which are standard protocols in the field. The research question and study selection criteria were defined in advance to avoid bias in the collection of literature. The PICO criteria used for selecting relevant studies are appropriate for the research question, and the eligibility criteria are clearly stated. The decision to exclude laboratory experiments and observational studies is also reasonable, given the limitations of these types of studies. The search databases used (PubMed, Web of Science, and Scopus) are commonly used in academic research, and the search formula was determined with the help of professional librarians. Additionally, the manual search of relevant literature cited by other articles and the list of literature that cited relevant reports is a good practice for ensuring completeness in the search.

Though, there are a few limitations to consider. The fact that the study is not registered on PROSPERO raises concerns about the potential for selective reporting or outcome reporting bias. Additionally, the search was limited to English-language papers, which may introduce language bias. Finally, there is no mention of the quality assessment of the included studies, which is an essential step in a meta-analysis.

Addtionally, Data extraction and quality assessment , lacks some details regarding the specific search strategies and databases used in the study. It would be beneficial to have more information on the specific keywords used in the search, the inclusion and exclusion criteria, and the search timeline. This information would help readers to understand the comprehensiveness of the search and the potential limitations of the study.

3. The discussion is of good quality as it explains the research findings and presents several possible explanations for the observed effects. The authors also compare their results to previous studies and discuss the potential implications of their findings. The use of figures to present data is also commendable.

However, there are some areas that could be improved. For instance, the discussion could have been more comprehensive by including a discussion on the limitations of the study, potential biases, and recommendations for future research. Additionally, while the authors have made a case for the possible mechanisms of action, they could have been more explicit in explaining how the observed effects could be attributed to the different toxic substances.

4. Conclusions seem to be supported by the findings reported in the study. The study quantitatively demonstrated the positive relationship between ventilation and task performance speed, which is an important finding. The statement that the findings will be helpful for policy decisions on indoor environmental management is also reasonable, as it suggests that increased attention should be given to ventilation rates in indoor environments.

However, the statement that the study demonstrates a positive effect of ventilation on health and intellectual productivity may be overly broad. While the study does demonstrate a positive effect of ventilation on task performance speed, it is unclear whether this translates to improvements in overall health or intellectual productivity in other areas. It is also worth noting that the study only examines the effect of ventilation on one specific task and may not generalise to other tasks or cognitive domains.

Finally, the statement that ventilation rates higher than the current standards might improve intellectual productivity is somewhat speculative. While the study does find a positive relationship between ventilation rate and task performance speed, it is unclear whether this relationship would hold at even higher ventilation rates, or whether there might be other negative consequences of very high ventilation rates (such as increased energy consumption or discomfort due to draftiness). More research would be needed to fully support this claim.

Author Response

Dear Mr. Reviewer 3,

I hope this email finds you well. I am extremely grateful for your invaluable assistance with the peer review process.

With regards to the feedback you have provided, I would like to address three specific points.

First, in regards to the quality assessment of the included studies in the meta-analysis in the Method section, this is outlined in line 123 of the manuscript.

Second, regarding the search strategy, this is described in the Supplementary Method section of the paper, and mentioned in lines 100-101 of the main manuscript. I have attached the Supplementary Method to this email for your convenience.

Finally, our findings suggest that the correlation between ventilation rate and task performance speed discussed in the conclusion is no longer observed beyond a limit of 10.7 L/s·person.

Thank you again for your invaluable feedback and please let me know if you require any further clarification.

Best regards,